# Adsorption Performance of Methylene Blue by KOH/FeCl_3_ Modified Biochar/Alginate Composite Beads Derived from Agricultural Waste

**DOI:** 10.3390/molecules28062507

**Published:** 2023-03-09

**Authors:** Heng Liu, Jiaqi Zhu, Qimei Li, Likun Li, Yanjun Huang, Yi Wang, Guozhi Fan, Lei Zhang

**Affiliations:** 1School of Chemistry and Environmental Engineering, Wuhan Polytechnic University, Wuhan 430023, China; 2China-Ukraine Institute of Welding, Guangdong Academy of Sciences, Guangzhou 510650, China

**Keywords:** corncob biochar, KOH/FeCl_3_ modified, methylene blue, adsorption performance, adsorption mechanism

## Abstract

In this study, high-performance modified biochar/alginate composite bead (MCB/ALG) adsorbents were prepared from recycled agricultural waste corncobs by a high-temperature pyrolysis and KOH/FeCl_3_ activation process. The prepared MCB/ALG beads were tested for the adsorption of methylene blue (MB) dye from wastewater. A variety of analytical methods, such as SEM, BET, FTIR and XRD, were used to investigate the structure and properties of the as-prepared adsorbents. The effects of solution pH, time, initial MB concentration and adsorption temperature on the adsorption performance of MCB/ALG beads were discussed in detail. The results showed that the adsorption equilibrium of MB dye was consistent with the Langmuir isothermal model and the pseudo-second-order kinetic model. The maximum adsorption capacity of MCB/ALG−1 could reach 1373.49 mg/g at 303 K. The thermodynamic studies implied endothermic and spontaneous properties of the adsorption system. This high adsorption performance of MCB/ALG was mainly attributed to pore filling, hydrogen bonding and electrostatic interactions. The regeneration experiments showed that the removal rate of MB could still reach 85% even after five cycles of experiments, indicating that MCB/ALG had good reusability and stability. These results suggested that a win-win strategy of applying agricultural waste to water remediation was feasible.

## 1. Introduction

In recent decades, chemical dyes have been widely used in textile, leather, paint and other industries, resulting in a large amount of dye wastewater [1]. Due to toxicity and poor biodegradability, it is essential that the dye wastewater is properly treated before being discharged into aquatic systems [2]. As a common cationic dye, methylene blue (MB) is widely used in print, leather and textile industries [3,4,5]. However, it has been proven to cause various health problems, such as vertigo, retching and eye burns [6]. Therefore, in recent years, it has been of great significance to remove MB from industrial wastewater [7].

Conventional treatment methods for MB include membrane separation, adsorption, electrochemical oxidation and the Fenton method [8,9,10,11]. Among various treatment technologies, the adsorption method is considered as the most promising method because of its good stability and efficiency [12]. Numerous adsorbents, such as graphene oxide, chitosan, activated carbon and bentonite, have been used to remove MB from aqueous solution [13,14,15,16]. However, the application of these adsorbents is limited by a high cost and restricted regeneration performance. Developing alternative adsorbents with the merits of low cost and eco-friendliness has always been pursued by scholars all over the world.

Biochar (BC) has attracted considerable attention as economical absorbents because of its large pore volume and facile modification [17,18,19]. Biochar is derived from agricultural and forestry wastes (e.g., peanut shells, corncob, rice husk, walnut shell, etc.) and prepared by pyrolysis under high temperature and under oxygen-free conditions [20,21,22]. In China, most corncobs are directly discarded or burned, resulting in resource waste and serious air pollution [23]. Corncob is a good raw material for the production of biochar to solve problems such as effective utilization of agricultural and forestry by-product wastes [24]. However, the biochar obtained by one-step direct pyrolysis usually has small adsorption capacity. In order to enhance its adsorption capacity, additional chemical modification is typically undertaken. KOH modification could improve the surface groups of biochar, thus improving the adsorption capacity of biochar [25]. Ma et al. [26] reported that the adsorption capacity of KOH modified corncob biochar for pollutants is six times higher than that of unmodified biochar. Furthermore, the modification of ferric chloride (FeCl_3_) can increase carbon yield and increase the number of functional groups of biochar, thus enhancing its adsorption performance [27]. However, the difficulties in cleaning and separating powdered biochar restrict its further industry application. Thus, the preparation of solid biochar is the key to the industrial application of biochar.

Sodium alginate (SA) is a kind of natural polysaccharide obtained mainly from marine brown algae, with rich hydroxyl and carboxyl functional groups in its polymer chain. It is widely used as the supporting material of various powder adsorbents in various methods, and shows a good application prospect in wastewater pollutant removal [28]. However, the prepared adsorbent has poor mechanical strength. To enhance the mechanical strength, glutaraldehyde is selected as the crosslinking agent. Sodium alginate composite corncob biochar pellets in the form of beads are deliberated in this research, which may overcome these disadvantages and are easy to reuse in batch studies. It is believed that the framework of KOH/FeCl_3_ modified corncob biochar loaded alginate matrix beads might show high selectivity towards the removal of MB from water.

Based on the hypothesis, the current work had the aim of fabricating facile, eco-friendly and biocompatible KOH/FeCl_3_ modified biochar/alginate composite beads for application in the adsorption of MB from water. The adsorbent was characterized by SEM, BET, XRD and FTIR. The effects of solution pH, initial concentration and contact time, adsorption temperature on the removal MB process were optimized. At last, the adsorption kinetics, isotherms and thermodynamics of adsorption of MB from water carried out to explore adsorption behavior.

## 2. Results and Discussion

### 2.1. Characterization

SEM images of the corncob biochar (CB), modified corncob biochar (MCB) and modified biochar/alginate composite beads (MCB/ALG) are shown in Figure 1. The surface of CB (Figure 1a) is relatively smooth, while the surface of MCB (Figure 1b) becomes rough, with more pore structures and more cracks. This is because the activation reaction between KOH and carbonaceous structure promotes the development of porous structure of porous carbon [29]. Obviously, this rich porosity may be conducive to improving the absorption performance of biochar. Although MCB/ALG (Figure 1c,e,g) has a spherical shape, its surface is relatively rough due to the irregular accumulation of biochar. It is worth noting that this phenomenon becomes more pronounced with the increase in MCB content in MCB/ALG. Due to the addition of large amounts of MCB, inhomogeneities can be observed on the surface of MCB/ALG−3.

To gain insight into the porous structure of the samples, N_2_ adsorption–desorption tests are performed using the isotherms and BJH pore size distribution curves shown in Figure 2. The porous structure parameters are summarized in Table 1. According to the IUPAC classification, the adsorption–desorption isotherms (Figure 2a) of three samples are considered to be type IV with H3 type hysteresis loop, implying the presence of porous structures in these samples. Compared to CB, the specific surface area of MCB is increased by more than twenty times. However, the poor pore structure of MCB/ALG compared to MCB suggests that some micropores and mesopores in MCB particles may be blocked after the composite process. Figure 2b shows the pore diameter distribution curves of the samples. It is observed that the pore diameters of three samples are mainly distributed around 8 nm, indicating that they have a mesoporous structure [30]. Table 1 shows that the modification leads to an increase in specific surface area and pore volume, and a decrease in pore diameter, indicating that the modification can induce a small porous structure in CB. In contrast, after the recombination process, the pore diameter of MCB/ALG increases slightly. In conclusion, the relatively high specific surface area and large pore diameter of MCB/ALG may be more conducive to the diffusion of contaminants into the internal pores and subsequent removal by adsorption.

Figure 3a shows the FTIR spectra of the samples to determine the various functional groups. The surface functional groups of corncob biochar were changed during the chemical modification of KOH and FeCl_3_ and high temperature carbonization process. All samples exhibit broad peaks around 3420 cm^−1^, indicating the presence of hydroxyl (O–H) stretching vibrations [31]. The increase in peak intensity of –OH after alkali modification compared to unmodified indicates an increase in oxygen-containing functional groups on the surface of biochar. These oxygen-containing functional groups enhance the adsorption of MB by providing active sites [32]. The peak around 1590 cm^−1^ and 1400 cm^−1^ are the aromatic ring (C=C) and C–N stretching vibrations, respectively. The peaks at 880 cm^−1^ and 810 cm^−1^ could be attributed to the C–H bending vibration outside the aromatic plane. It has been reported that the occurrence of aromatization processes during the modification process may lead to the enhancement of the intensity of the above-mentioned peaks. It can be seen that the prepared MCB/ALG inherits the functional groups of MCB and SA. It proves that the composite beads were successfully synthesized. The XRD spectra of the samples are illustrated in Figure 3b. For the four samples in the figure, the broad diffraction peaks near 2θ = 22° are characteristic peaks for cellulose and hemicellulose [33]. After modification, FeCl_3_ significantly changed the XRD spectrum of CB. Both MCB and MCB/ALG−1 samples show a wide diffraction peak at a 2θ value of 43.2°, belonging to the plane of graphite carbon. The diffraction peaks of Fe_3_O_4_ (2θ = 30.2°, 43.2°, 57.2°, 62.7°) and γ-Fe_2_O_3_ (2θ = 35.5°) can be clearly observed on MCB and MCB/ALG [34]. This is mainly due to the addition of FeCl_3_ in the modification process, which introduces a small amount of Fe. The results demonstrate the presence of graphene structures in MCB and MCB/ALG, which facilitates the adsorption of MB [35].

### 2.2. Effect of pH on Adsorption

As an important factor in the adsorption process, the initial pH value can affect the charge change on the adsorbent surface, thus affecting the adsorption efficiency of MB [36]. The effect of pH on MB adsorption from water on adsorbents is presented in Figure 4a. From Figure 4a, it can be seen that the removal efficiencies of MB have changed slightly in the pH value range of 3.0–10.0. This suggests that the adsorption process is less affected by the electrostatic force between MB and adsorbent under the solution pH value of 3.0–7.0, and the adsorption process may be dominated by pore filling and hydrogen bonding. As the solution pH value enhances, the removal efficiency of MB increased significantly. Therefore, it is not conducive for the adsorption of MB on MCB/ALG samples under acidic conditions. Compared with three MCB/ALG adsorbents, MCB/ALG−1 shows a better adsorption performance. The affinity of biochar to MB gradually increases with increasing pH. The solution has a strong electrostatic attraction between the positively charged biochar adsorbent and the negatively charged MB. As shown in Figure 4b, the adsorbent surface shows a more negative charge density at higher pH values. The surface of biochar contains chemical groups, such as –OH and –COOH, and the pH of the dye changes its charge. At acidic conditions, the capacity of the binding sites on biochar is limited. At low pH conditions, MB is a positively charged cationic dye, which leads to electrostatic repulsion between MB and biochar, thereby reducing the removal rate. Through later analysis, the study found that electrostatic attraction is not the only adsorption method [37]. In addition to the adsorption of surface functional groups, it also includes intra-particle diffusion. More adsorption mechanisms have been explored later. Hence, increasing the pH of the solution after reaching the adsorption equilibrium does not significantly reduce the removal rate. When the pH value increased to 10, the absolute value of ΔpH of biochar decreased, showing a slight decrease in removal rate. So, exploring suitable adsorption conditions is conducive to the maximization of adsorption efficiency.

### 2.3. Adsorption Kinetics

Figure 5a shows the time evolution of MB adsorption on MCB/ALG−1 and MCB. In order to evaluate the adsorption kinetic behavior of MB on adsorbent, pseudo-first-order, pseudo-second-order and intra-particle diffusion models were used to fit the experimental data [38].

The pseudo-first-order kinetic model:(1)$$q_{t}=q_{e}[1-e^{-k_{1}t}]$$

The pseudo-second-order kinetic model:(2)$$q_{t}=\frac{q_{e}^{2}k_{2}t}{1+k_{2}q_{e}t}$$

The intra-particle diffusion kinetic model:(3)$$q_{t}=k_{i}t^{1/2}+C$$
where *q_t_* (mg/g) represents the amount of MB adsorbed by a mass unit of adsorbent at a predetermined time *t* (min), whereas *k*_1_ (/min) and *k*_2_ (g/mg·min) are the rate constants of the pseudo-first-order and pseudo-second-order kinetic models, respectively. *k_i_* (mg/g·min^1/2^) is the rate constant for the intra-particle diffusion model, and *C* (mg/g) is a constant that represents the boundary layer thickness.

In this work, the kinetic parameters were obtained by using non-linear fitting method, and the applicability of adsorption kinetic model was determined by the root mean squared error (*RMSE*). In principle, the lower the *RMSE* value, the more suitable the model fits. *RMSE* could be calculated using the following Equation:(4)$$RMSE=\sqrt{\frac{1}{n}\sum_{i=1}^{n}\left(q_{cali}-q_{\exp i}\right)}$$
where *q_cali_* and *q_expi_* are the predicted and measured values of the adsorption capacity at time *t*, respectively, and *n* is the number of experimental data.

As shown in Figure 5a, the adsorption process can be divided into three stages: the fast adsorption stage, the slow adsorption stage, and the stabilization stage. During the fast adsorption stage, about 90% of the adsorption occurred within 250 min, which may be because the high concentration of MB at the interface between the adsorbents and the solution promoted a large mass transfer driving force, resulting in MB rapidly occupying the adsorption site. In addition, the values of kinetic parameters and their *RMSE* values are summarized in Table 2. As a key parameter, the root mean square error of the pseudo-second-order model is lower than that of the pseudo-first-order model. In contrast, the fitted curves of the pseudo-second-order model are always closer to the experimental data points, indicating that the pseudo-second-order model can better describe the adsorption kinetics. These results show that the pseudo-second-order model is more suitable for describing the adsorption process than the pseudo-first-order model, which indicates that the rate–determining step of MCB/ALG−1 may be a chemisorption process.

The intra-particle diffusion model is further applied to determine the rate-controlling step of the adsorption processes. According to Figure 5b, the plots of *q_t_* versus *t*^0.5^ are comprised of three linear segments and do not pass through the origin. It is proved that the adsorption process of MB on MCB/ALG−1 and MCB is multi-step, and intra-particle diffusion is not the only rate-limiting step [39]. In the first stage, the fastest adsorption rate (maximum slope) is associated with film diffusion, where MB molecules migrate from solution to the outer surface of MCB/ALG−1 and MCB, as the large concentration gradient provides sufficient driving force. The second stage has a relatively high adsorption rate and shows a progressive adsorption phase corresponding to the intra-particle diffusion of MB molecules through the internal pores and cavities of the adsorbent. In the third stage, the diffusion rate in the pores is further reduced and the adsorption gradually reaches equilibrium. Compared to the rate constants of the stages, the rate constants of the first stage are much higher than those of the other stages, which implies that the film diffusion is the dominant rate limiting step in the whole process; while the linear part near the platform in the late stage, indicating that intra-particle diffusion is not the only rate-limiting factor, the adsorption rate is also affected and controlled by the external diffusion step and the surface diffusion of the adsorbent [36,40].

### 2.4. Adsorption Isotherm

The adsorption isotherm curve is helpful to analyze the interaction between the adsorbent and the adsorbate and the characteristics of the adsorption layer. Langmuir and Freundlich isotherm models are used to describe the adsorption data of MB on MCB/ALG−1 samples at 303, 313 and 323 K. The Langmuir and Freundlich isotherm models can be expressed as follows [41].

The Langmuir isothermal model:(5)$$q_{e}=\frac{q_{m}K_{L}C_{e}}{1+K_{L}C_{e}}$$

The Freundlich isothermal model:(6)$$q_{e}=K_{F}C_{e}^{1/n_{F}}$$
where *q_e_* and *q_m_* (mg/g) represent the equilibrium adsorption amount and the maximum adsorption amount, respectively; *C_e_* (mg/L) represents the equilibrium adsorption concentration; *K_L_* is the Langmuir constant which is related to the affinity of binding sites (L/mg). *K_F_* is the Freundlich isothermal constant and *n_F_* is the heterogeneity factors.

The characteristic constants of Langmuir and Freundlich isotherm models are shown in Table 3, and the curves are shown in Figure 6. The Langmuir isothermal model assumes that monolayer adsorption occurs on the adsorbent surface with equivalent adsorption sites, while the Freundlich isothermal model is used to describe equilibrium data and adsorption characteristics for a heterogeneous surface. The Langmuir model has a lower root mean square error compared to the Freundlich model, indicating that the Langmuir isotherm model is a better fit than the Freundlich isotherm model for the adsorption data in the current experiments, in agreement with a previous report [42]. Thus, the adsorption of MB on MCB/ALG−1 samples is consistent with monolayer adsorption. In addition, an increasing trend of MB adsorption is observed in the temperature range of 303–323 K, clearly indicating its endothermic nature. Meanwhile, the parameter n_F_ of the Freundlich model is greater than 1, indicating a strong affinity between the MCB/ALG−1 samples and the MB molecule, which is a favorable adsorption process. The maximum adsorption capacities (*q_m_*) of MCB/ALG−1 calculated according to the Langmuir model were 1373.49, 1457.28 and 1485.03 mg/g at 303, 313 and 323 K, respectively. Table 4 lists the maximum adsorption capacities (*q_m_*) for MB on biochar derived from various waste reported in previous studies. It is noted that the adsorption capacity of MCB/ALG−1 is superior or comparable to that of most adsorbents reported in the literature.

### 2.5. Adsorption Thermodynamics

Temperature is an important parameter to control species adsorption in the system. The Gibbs free energy change (Δ*G*), enthalpy change (Δ*H*) and entropy change (Δ*S*) play a key role in determining the heat exchange and spontaneity of the adsorption process. Application of the flow equation to calculate thermodynamic parameters [50].
(7)$$\Delta G=-RT\ln K_{d}$$
(8)ΔG=ΔH−TΔS
(9)$$\ln K_{d}=\frac{\Delta S}{R}-\frac{\Delta H}{RT}$$
where *K_d_* is the thermodynamic equilibrium constant; *R* (8.314 J/(mol·K)) stands for the gas constant; *T* (K) represents the experimental temperature; Δ*S* (J/(mol·K)) represents the entropy change of the system; Δ*H* (kJ/mol) represents the enthalpy change and Δ*G* (kJ/mol) represents the Gibbs free energy.

The calculation results of thermodynamic equilibrium coefficient are shown in Table 5. The enthalpy of the adsorption process become positive (Δ*H* > 0) at different temperatures, indicating that the adsorption of MB on MCB/ALG−1 sample is a heat absorption reaction and the increase in temperature favors the adsorption process, which is consistent with the isothermal model analysis. The change in entropy is also positive (Δ*S* > 0), proving that the adsorption process is irreversible and proceeds along the direction of increasing system disorder. Meanwhile, the Δ*G* values are negative, thus indicating that the adsorption process was spontaneous. Hence, the MB adsorption on MCB/ALG−1 is endothermic and spontaneous.

### 2.6. Regeneration and Reusability

In practical applications, it is necessary to perform adsorption–desorption tests on the adsorbent, since the reusability of the adsorbent is a key factor for economic results [51]. From Figure 7, it can be found that the MCB/ALG−1 sample can still reach 85% MB removal after five adsorption–desorption cycles. Therefore, the MCB/ALG−1 sample is a promising adsorbent with good reusability.

## 3. Materials and Methods

### 3.1. Materials

The corncobs were obtained from the countryside of Wuhan, Hubei province, China. Methylene blue, ferric chloride hexahydrate (FeCl_3_·6H_2_O, AR, 98%), potassium hydroxide (KOH, GR, 85%), hydrochloric acid (HCl, 36%), glutaraldehyde (25%), calcium chloride (CaCl_2_, AR, 96%) and sodium alginate (200 ± 20 mPa s), sodium chloride (NaCl, AR, 99.5%), nitric acid (HNO_3_), sodium hydroxide (NaOH, AR, 95%) were supplied from Sinopharm Chemical Reagent Co., Ltd (Shanghai, China). All reagents are analytical grade and used directly. All water mentioned in this paper was deionized water (18.2 MΩ cm).

### 3.2. Preparation of KOH/FeCl_3_ Modified Corncob Biochar (MCB)

To prepare KOH/FeCl_3_ modified corncob biochar (MCB), the corncob was washed with deionized water 3 times to remove impurities on its surface. After that, the corncob powder and KOH (mass ratio of corncob powder: KOH = 1:1.87) were mixed with 100 mL 2.5 mol/L FeCl_3_·6H_2_O for modification [35]. The mixture was stirred in a 60 °C water bath for 2 h and maintained overnight in an oven at 60 °C. Subsequently, the corncob powder was pyrolyzed in a tubular furnace at 800 °C for 2 h at a heating rate of 10 °C/min [51]. During the pyrolysis process, N_2_ was used as a protective gas at a flow rate of 200 mL/min. Finally, after cooling to room temperature, the biochar was washed with 1 mol/L HCl, and then with distilled water at room temperature up to neutral pH. The final sample was dried in an oven at 60 °C. The resulting modified biochar was denoted by MCB, and the unmodified biochar was denoted as CB.

### 3.3. Preparation of MCB/ALG Composite Beads

Generally, 0.2 g SA was dissolved in 10 mL deionized water. MCB (0.2, 0.4, 0.6 g) was added, and the mixture was stirred to obtain a homogeneous system. Then, the above mixture was poured slowly into a syringe (1 mL) and was injected into 1% CaCl_2_ solution at a uniform rate for crosslinking of 12 h. The beads were rinsed repeatedly with deionized water to remove Ca^2+^ and residual MCB particles. Subsequently, the obtained beads were immersed into a mixture solution containing 1 wt% glutaraldehyde for 12 h with continuously shaking and were rinsed 5–6 times with deionized water. The resulting sample was denoted MCB/ALG−1, MCB/ALG−2 and MCB/ALG−3, according to the mass ratio of MCB and SA in the modified biochar/alginate composite beads. Finally, the prepared MCB/ALG were freeze-dried. Figure 8 illustrates the process of MCB/ALG preparation.

### 3.4. Characteristics of Samples

The surface morphology and particle size of adsorbents were studied by scanning electron microscopy (SEM, Hitachi, S-3000N, Tokyo, Japan). Fourier transform infrared spectroscopy (FTIR, Thermo Scientific, Nicolet IS10, Waltham, MA, USA) recorded chemical constituents of adsorbents in the range from 4000 to 400 cm^−1^. The specific surface area, pore size and pore volume of samples were measured by the Brunauer–Emmet–Teller (BET, Micromeritics, ASAP2460, Atlanta, GA, USA). The crystalline state of the adsorbents was analyzed by X-ray diffraction (XRD, Shimadzu, XRD-7000, Kyoto, Japan). The point of zero charge (pH_PZC_) for MCB-ALG beads was measured by the batch equilibrium method. The pH values of 0.1 and 0.01 mol/L NaCl solutions were adjusted from 3.0 to 12.0 by adding HNO_3_ or NaOH solution. The solution was puffed with nitrogen at room temperature to remove dissolved carbon dioxide until the initial pH (pH_i_) was stable. Then, 0.1 g of MCB/ALG−1 was introduced into 50 mL, respectively, and the suspension was shaken for 24 h. The final pH values (pH_f_) of supernatant were recorded. The difference between the initial and final pH values (ΔpH = pH_f_ − pH_i_) was plotted against the pH_i_. The point of intersection of the resulting curve at which ΔpH = 0 was the pH_PZC_.

### 3.5. Batch Adsorption Experiments

The stock solution of MB dye with a concentration of 10,000 mg/L was prepared and was diluted to obtain the MB solution required for the experiment. Each sample was tested three times under the same conditions, and the average value of the results was taken.

Ten milligrams of adsorbent (MCB, MCB/ALG) was placed into a centrifuge tube containing 10 mL MB solution and agitated at 170 rpm on a shaker. After stirring at 25 °C for 24 h, the supernatant was separated by a 0.45 μm membrane filter. The concentration of MB in the supernatant was measured by UV–visible spectrophotometer (Beijing general analytical Instrument, T6, China) at a wavelength of 664 nm. The adsorption capacity (*q_e_*, mg/g) and removal efficiency (R, %) were calculated according to Equations (10) and (11).
(10)$$q_{e}=\frac{(C_{0}-C_{e})V}{m}$$
(11)$$R=\frac{\left(C_{0}-C_{e}\right)}{C_{0}}\times 100\%$$
where *C*_0_ and *C_e_* are the initial and equilibrium concentrations of MB, respectively, *m* (g) represents the dosage of adsorbent, *V* (L) is the MB solution volume.

In order to study the effect of pH value on the removal efficiency of MB, the pH value of solution was changed from 3 to 12. The initial pH value of the MB solution measured by a pH meter and adjusted by using 0.1 mol/L HNO_3_ and NaOH (aq). For adsorption kinetics experiments, the MB solution with a concentration of 1200 mg/L was shaken at 25 °C for a range of 10 to 2000 min. Adsorption isotherm experiments was tested at the MB solution concentration of ranging from 25 to 4000 mg/L at 303, 313 and 323 K. The thermodynamic studies were conducted at different temperatures under the same experimental conditions.

### 3.6. Regeneration Experiment

For desorption and regeneration studies, experiments were performed in 10 mL of MB solution (1200 mol/L) with 10 mg MCB/ALG−1 and shaken at 25 °C for 24 h. After saturation, the spent MCB/ALG−1 beads were washed with 50 mL solution containing 1 mol/L HNO_3_ for 12 h. Then, MCB/ALG−1 beads were washed thoroughly with deionized water and dried at 40 °C. The adsorption–desorption experiment was repeated five times.

## 4. Conclusions

In this study, biochar/alginate composite beads from waste corncobs were successfully fabricated by integrating a pyrolysis process and KOH/FeCl_3_ modification. By comparing the properties and adsorption performance of MCB/ALG with different KOH modified ratios, it was found that the MCB/ALG−1 sample had the most graphitized structure and the largest adsorption capacity, with maximum adsorption capacities of 1373.49 mg/g at 303 K. The adsorption followed the pseudo-second-order kinetic model and the Langmuir isotherm model. Both the intra-particle diffusion model and the liquid film diffusion were involved in the adsorption process, and the film diffusion was the main rate-limiting step. The negative values of Δ*G* and the positive value of Δ*H* confirmed the spontaneous and endothermic nature of MB adsorption on MCB/ALG−1. In addition, MCB/ALG−1 showed the excellent adsorption capacities and considerable reusability. Therefore, MCB/ALG−1 can be used as a promising adsorbent material towards the treatment of MB dye-polluted water or wastewater.

## Figures and Tables

**Figure 1 molecules-28-02507-f001:**
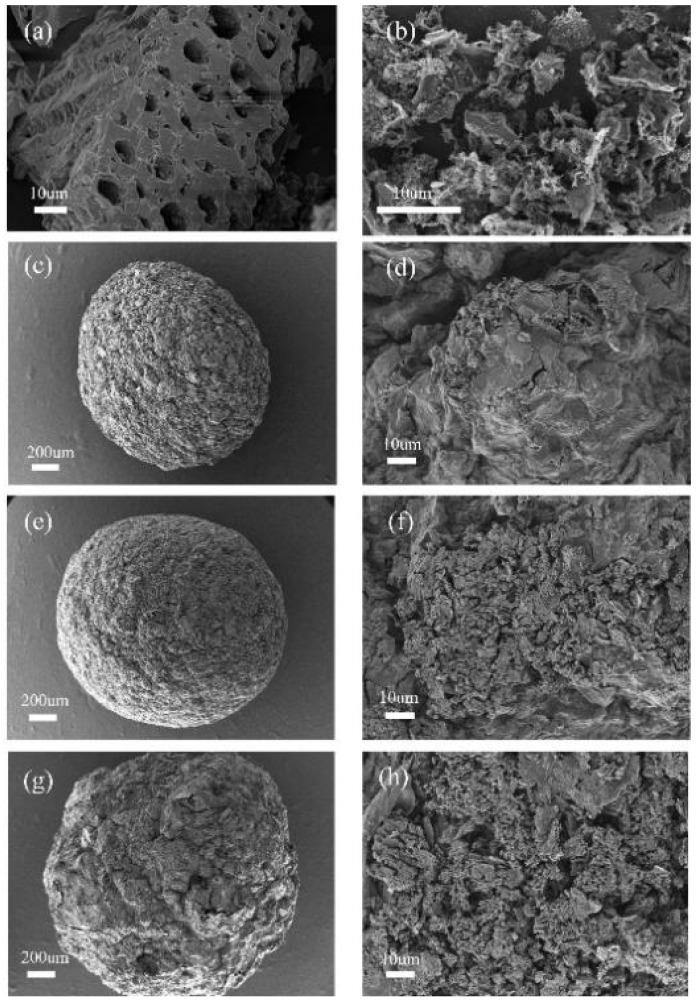
SEM images of (**a**) CB, (**b**) MCB, (**c**,**d**) MCB/ALG−1, (**e**,**f**) MCB/ALG−2, (**g**,**h**) MCB/ALG−3.

**Figure 2 molecules-28-02507-f002:**
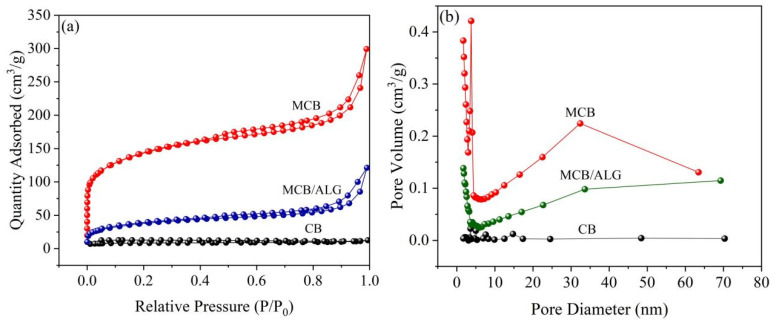
(**a**) N_2_ adsorption–desorption isotherms and (**b**) the pore size distribution of BC samples.

**Figure 3 molecules-28-02507-f003:**
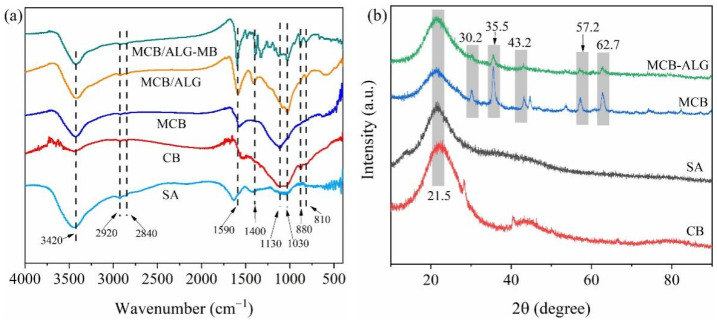
(**a**) FTIR and (**b**) XRD spectra of samples.

**Figure 4 molecules-28-02507-f004:**
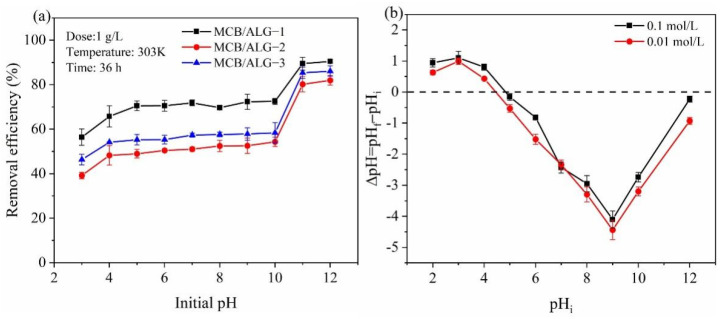
Effect of pH on MB by MCB/ALG−1, MCB/ALG−2 and MCB/ALG−3: (**a**) removal efficiency; (**b**) determination of pH_PZC_ of MCB/ALG−1 in NaCl solutions by batch equilibrium.

**Figure 5 molecules-28-02507-f005:**
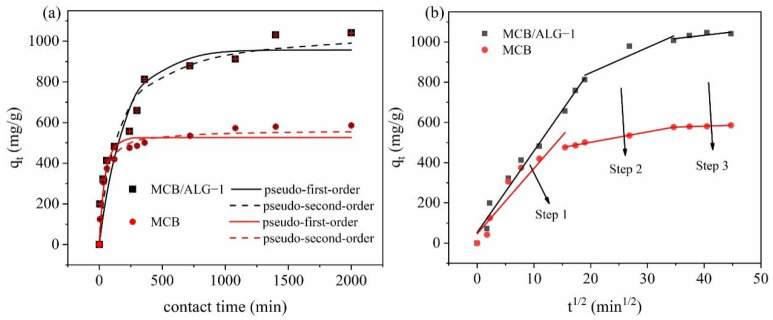
(**a**) Kinetics fitting results of MB adsorption on MCB/ALG−1 and MCB; (**b**) intra-particle diffusion model for MB uptake onto MCB/ALG−1 and MCB.

**Figure 6 molecules-28-02507-f006:**
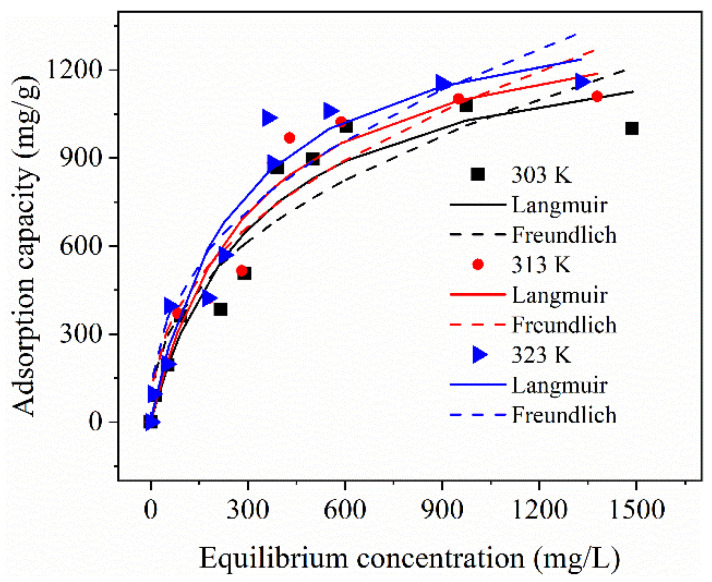
Adsorption isotherms of MB on MCB/ALG−1 sample at different temperature.

**Figure 7 molecules-28-02507-f007:**
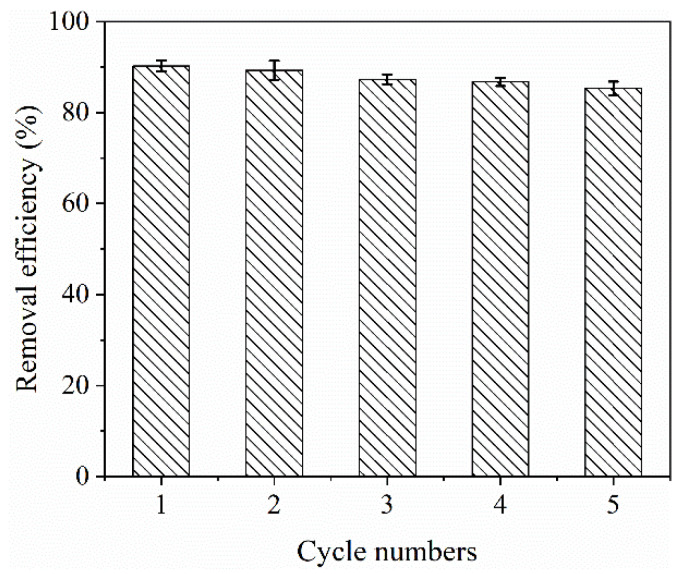
Removal efficiency of the regeneration cycle.

**Figure 8 molecules-28-02507-f008:**
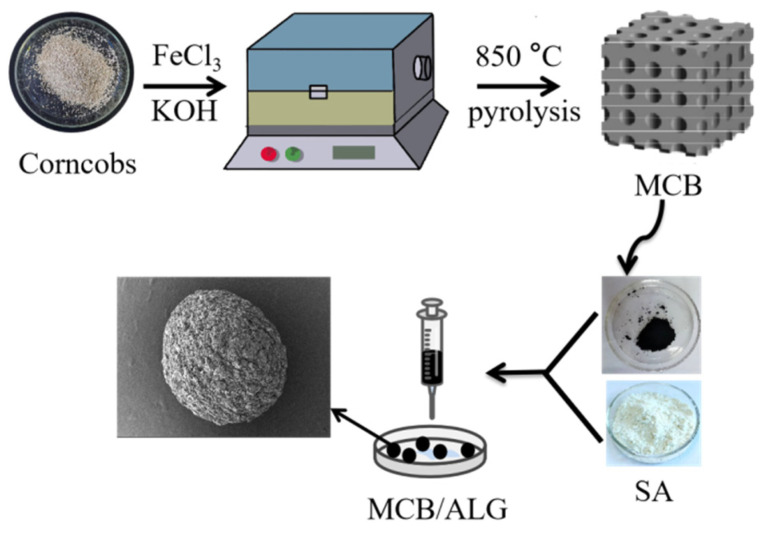
Schematic illustration of the formation of MCB/ALG.

**Table 1 molecules-28-02507-t001:** BET parameters of CB, MCB and MCB/ALG.

Sample	BET Surface Area (m^2^/g)	Pore Volume (cm^3^/g)	Pore Diameter (nm)
CB	25.9	0.006	7.6
MCB	468.4	0.3	6.0
MCB/ALG	128.8	0.1	8.1

**Table 2 molecules-28-02507-t002:** Kinetic parameters for the adsorption of MB on MCB/ALG−1 and MCB.

Sample	*q_e,exp_* (mg/L)	Pseudo-First-Order Model			Pseudo-Second-Order Model		
		*q_e,cal_* (mg/g)	*k*_1_ (/min)	*RMSE*	*q_e,cal_* (mg/g)	*k*_2_ (g/mg·min)	*RMSE*
MCB/ALG−1	1040.7	957.3	0.005	105.1	1055.2	7.31 × 10^−6^	81.4
MCB	585.92	525.8	0.02	46.0	562.7	6.09 × 10^−5^	25.1

**Table 3 molecules-28-02507-t003:** Adsorption parameters obtained from Langmuir and Freundlich isotherm models.

T (K)	Langmuir			Freundlich		
	*q_m_* (mg/g)	*K_L_* (L/mg)	*RMSE*	*K_F_* (L/mg)	*n_F_*	*RMSE*
303	1373.49	0.0030	105.05	55.93	2.37	127.88
313	1457.28	0.0031	94.47	59.26	2.3	116.74
323	1485.03	0.0037	106.91	70.73	2.45	127.25

**Table 4 molecules-28-02507-t004:** Comparison of various biochar materials for MB removal.

Raw	Pyrolysis Temperature (°C)	*q_m_* (mg/g)	Reference
Tamarind seed	500	102.77	[43]
Sodium carboxymethyl cellulose	900	249.6	[44]
Bamboo	600	286.1	[45]
Alfalfa	600	326.90	[35]
Rattan stalks	600	359	[46]
Coffee grounds	600	367	[47]
Waste tea	450	683.6	[48]
Corncobs	800	1373.49	This work
Corncob-to-xylose residue	850	1563.9	[49]

**Table 5 molecules-28-02507-t005:** Thermodynamic parameters for adsorption of MB on MCB/ALG−1.

T (K)	Δ*G* (kJ·mol ^−1^)	Δ*H* (kJ·mol ^−1^)	Δ*S* (kJ·mol ^−1^ K^−1^)
303	−0.2551	9.4163	0.0754
313	−0.3726		
323	−0.6369		

## Data Availability

Compound data sets are publicly available. Samples are available from the authors.

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
