# Peer review of "Adsorption Performance of Methylene Blue by KOH/FeCl3 Modified Biochar/Alginate Composite Beads Derived from Agricultural Waste"

_molecules, 2023, doi:10.3390/molecules28062507_

Round 1
Reviewer 1 Report
In this study, the authors reported a simple, eco-friendly and biocompatible KOH/FeCl3 modified biochar/alginate composite bead, which was used to adsorb methyl bromide in water. MCB/ALG-1 can be used as a promising adsorption material to treat MB dye contaminated water or wastewater. However, this work needs to be revised before final publication.
(1) In abstract, it is mentioned that the high adsorption performance of MCB/ALG is mainly attributed to pore filling, hydrogen bonding and electrostatic interaction, but there is no specific description of hydrogen bonding in subsequent relevant studies.
(2) Table 1, the significant digit should be ground, actually, it can be accurate to one digit.
(3) Some recent articles on biochar should be reviewed, such as Carbon Research 1, 8 (2022) and Carbon Research 1, 4 (2022).
(4) A small amount of Fe was introduced during the modification process. How did Fe3O4 generate? More characterizations were needed. Again, what were the roles of these Fe minerals in promoting the adsorption ability for MB.
(5) Fig. 4, MCB or MCB/ALG-1? From the description, it should be MCB/ALG-1 while it was read MCB in the figure.
Author Response
Q1. In abstract, it is mentioned that the high adsorption performance of MCB/ALG is mainly attributed to pore filling, hydrogen bonding and electrostatic interaction, but there is no specific description of hydrogen bonding in subsequent relevant studies.
Reply: As we know, the oxygen-containing groups were introduced onto the biochar through potassium activator at higher carbonization temperature. As we described in Line 117-120 and Line 141-146, this suggests that the adsorption process is less affected by the electrostatic force between MB and adsorbent under the solution pH value of 3.0–10.0, and the adsorption process may be dominated by pore filling, hydrogen bonding. Thanks!
Q2. Table 1, the significant digit should be ground, actually, it can be accurate to one digit.
Reply: Following the reviewer’s comment, the significant digit is accurate to one digit. The revised manuscript has been corrected in Table 1 and highlighted in yellow. Thanks!
Q3. Some recent articles on biochar should be reviewed, such as Carbon Research 1, 8 (2022) and Carbon Research 1, 4 (2022).
Reply: Following the reviewer’s comment, the related articles on biochar have been cited in this article, see references 19 and 22. Thanks!
Chen, H., Gao, Y., Li, J., Fang, Z., Bolan, N., Bhatnagar, A., Gao, B., Hou, D., Wang, S., Song, H., Yang, X., Shaheen, S.M., Meng, J., Chen, W., Rinklebe, J. and Wang, H. (2022) Engineered biochar for environmental decontamination in aquatic and soil systems: a review. Carbon Research. 2022,1:4.
Liu, Z., Xu, Z., Xu, L., Buyong, F., Chay, T.C., Li, Z., Cai, Y., Hu, B., Zhu, Y. and Wang, X. (2022) corrected biochar: synthesis and mechanism for removal of environmental heavy metals. Carbon Research, 2022,1:8.
Q4. A small amount of Fe was introduced during the modification process. How did Fe3O4 generate? More characterizations were needed. Again, what were the roles of these Fe minerals in promoting the adsorption ability for MB.
Reply: the generation of Fe3O4 can be listed as follows:
FeCl3 can significantly reduce graphitization temperature and increase carbon yield, and the graphitization structure can enhance the interactions between the adsorbent and MB(see line 54 and 128). After modification, FeCl3 significantly changed the XRD spectrum of CB. Both MCB and MCB/ALG-1 samples showed a wide diffraction peak at a 2θ value of 43.2°, which may be attributed to the plane of graphite carbon, and facilitated the adsorption of MB.
Q5. Fig. 4, MCB or MCB/ALG-1? From the description, it should be MCB/ALG-1 while it was read MCB in the figure.
Reply: Following the reviewer’s comment, we have checked the error , and Fig. 4a is MCB/ALG-1, MCB/ALG-2 and MCB/ALG-3, respectively. Fig 4b is MCB/ALG-1. The revised manuscript has been corrected in Fig. 4 and highlighted in yellow. Thanks!
Thanks for your comments and suggestions on our manuscript!
Reviewer 2 Report
MS ID: molecules-2261888-peer-review-v1
MS title: Adsorption performance of methylene blue by KOH/FeCl3 modified biochar/alginate composite beads derived from agricultural waste
Review report:
In this manuscript, authors have prepared new composites based on modified biochar/alginate for methylene blue removal from wastewater. The manuscript sound good, well-structured and written. Therefore, it can be accepted after addressing the following major comments.
Important comments.
- Thera are many typos in the manuscripts.
- Materials and methods: please describe the role for all materials used for preparation of the composites. For example, the crosslinkers CaCl2 and glutaraldehyde (note: without crosslinking, NCs are not stable and cannot be used as adsorbents). Please confirm the crosslinking of the alginate-containing adsorbents.
- The effect of pH value in the preparation process. The reason behind selection of precursors ratio. Cite the necessary work(s). Is the use of syringe being part of obtaining beads? You did pyrolyze at 850 degree Celsius, is this temperature standard or is it optimized or used according to similar work already reported. Please enrich the preparation section with the clear steps and information that benifit researchers.
- As the experiments were performed in triplicate, please include error bars to the results wherever possible.
- Figure 4a. is it for MCB1,2,3 or MCB/ALG-1,2,3?
- Cost and availability of the adsorbent must be discussed, in brief, in the introduction and/or in the last section as part of Table 4. Compare with literatures and with commercial adsorbents.
-
Additional few minor corrections:
- Add “and” at the end series at lines 32, 43.
- Check the comma type at lines 78 and 326.
- Line 89: (b) MCB850. Is it differ from MCB? Otherwise no need to use two terms for one item. Remove 850 if it is the same as MCB.
- Consider minus sign as (−), which can be found as (U+2212). Correct it everywhere in the manuscript like at lines 113, 119, 120, 255-256 and so on.
- FeCl3 (subscript) not FeCl3, e.g., in lines 130, 350, etc. also, CaCl2 line 299.
- The range of pH described in line 140 not match the figure range 3–10, and the change here is not significant (lines 140-141).
- The unit of ki: no minus, so revise the units in the whole manuscript.
- Line 175: non-linear.
- Figure 6. Correct freundich to Freundlich.
- Line: nF (F is a subscript).
- Lines 208-210: the statement may need more discussion and to be supported with reference(s).
- Line 243: correct English (It shows).
- In table 4: capitalize the first letter (tamarind, corncobs).
- Line 251: Kd instead of K.
- Section 3.1.: consider reporting purity for all chemicals.
- Line 285: referring to KOH mass make no sense because its purity was not reported.
- What is the idea behind selection of the NC precursors ratio as corncon:KOH: Fe3+ of 1:1.87:2.5 M Fe3+. Is it based on any criteria or is it optimized first or performed according to a previous, justified method?
- Line 288: report the N2 gas flow rate.
- Line 311: pore volume of …(of what?).
- Lines 343-344: the sentence is not clear.
- Line 344: 25°C or 25 °C, keep one style.
- Line 345: add space (to be 12 h).
- Line 345: it is important to report the HNO3 volume used in each cycle.
- Used one style to abbreviate composite. Is MCB-ALG differs from MCB-ALG.
- Line 376: no XPS test in the report?
Author Response
Reviewer #2:
Q1. Thera are many typos in the manuscripts.
Reply: Following the reviewer’s comment, we have checked carefully , and typos in the manuscripts have been revised in the manuscript and and highlighted in yellow. Thanks.
Q2. Materials and methods: please describe the role for all materials used for reparation of the composites. For example, the crosslinkers CaCl2 and glutaraldehyde (note: without crosslinking, NCs are not stable and cannot be used as adsorbents). Please confirm the crosslinking of the alginate-containing adsorbents.
Reply: In fact, the roles of all materials used for preparation of the composites are described in the background (See introduction section, line 47-69). thanks.
Q3. The effect of pH value in the preparation process. The reason behind selection of precursors ratio. Cite the necessary work(s). Is the use of syringe being part of obtaining beads? You did pyrolyze at 850 degree Celsius, is this temperature standard or is it optimized or used according to similar work already reported. Please enrich the preparation section with the clear steps and information that benifit researchers.
Reply: Following the reviewer’s comment, we have enriched the relevant literature[ ref.35, 51]. According to the relevant literature [51] and the previous optimization work, 850℃ was selected in this paper. The references are highlighted in yellow. In addition, The effect of pH value in the preparation process is not involved in this paper, and the syringe is used to obtain MCB/ALG pellets.
Q4. As the experiments were performed in triplicate, please include error bars to the results wherever possible.
Reply: Following the reviewer’s comment, error bar has been used in the revised manuscript.
Q5. Figure 4a. is it for MCB1,2,3 or MCB/ALG-1,2,3?
Reply: Following the reviewer’s comment, we have checked the error , and Fig. 4a is MCB/ALG-1, MCB/ALG-2 and MCB/ALG-3, respectively. Fig 4b is MCB/ALG-1. The revised manuscript has been corrected in Fig. 4 and highlighted in yellow. Thanks!
Q6: Cost and availability of the adsorbent must be discussed, in brief, in the introduction and/or in the last section as part of Table 4. Compare with literatures and with commercial adsorbents.
Reply: Cost and availability are crucial for the adsorbent. At present, the adsorbent raw materials commonly used in commercialization include plant raw materials (such as walnut shell, charcoal, coconut shell, bamboo, bark, fruit core, rice shell, oil palm shell, etc.), coal raw materials (such as anthracite, weakly sticky coal, lignite and peat, etc.), and petroleum raw materials (such as asphalt, petroleum coke, petroleum residue, etc.); Plastics (such as polyvinyl chloride, polypropylene, phenolic resin, urea-formaldehyde resin, polytetrafluoroethylene, etc.) and others, such as animal bones, molasses, old tires, etc. Activated carbon is usually prepared by chemical activation.
Biochar and sodium alginate, as the two most common and abundant raw materials, it is very available. the current work has targeted to fabricate a facile, eco-friendly and biocompatible KOH/FeCl3 corrected biochar/alginate composite beads and applied for the adsorption of MB from water. Therefore, we only compared the adsorption performance without considering the cost factor, as shown in Table 4. It is difficult to compare the cost factors because the prices of raw materials in different regions are different. This study is based on the local rich natural resources, which are cheap and available. It overcomes the shortcomings of adsorbents reported in the previous literature and shows superior or comparable to that of most adsorbents reported in the literature ( see Table 4). The revised manuscript has highlighted the adsorption capacity (Line 253-255). Thanks.
Q7: Add “and” at the end series at lines 32, 43.
Reply: Related errors have been corrected in the revised manuscript, and highlighted in yellow.
Q8: Check the comma type at lines 78 and 326.
Reply: Related comma type have been corrected in revised manuscript.
Q9: Line 89: (b) MCB850. Is it differ from MCB? Otherwise no need to use two terms for one item. Remove 850 if it is the same as MCB.
Reply: Thanks the reviewer’s comment, we have checked carefully, the error in the manuscript has been revised and and highlighted in yellow (see Fig.1, Line 91). Thanks.
Q10: Consider minus sign as (−), which can be found as (U+2212). Correct it everywhere in the manuscript like at lines 113, 119, 120, 255-256 and so on.
Reply: Thanks the reviewer’s comment, we have checked carefully, the errors in the manuscript have been revised and and highlighted in yellow . Thanks.
Q11: FeCl3 (subscript) not FeCl3, e.g., in lines 130, 350, etc. also, CaCl2 line 299.
Reply: Thanks the reviewer’s comment, the related errors have been corrected in the revised manuscript.
Q12: The range of pH described in line 140 not match the figure range 3–10, and the change here is not significant (lines 140-141).
Reply: Thanks the reviewer’s comment, the related errors have been corrected in the revised manuscript and highlighted in yellow (Line 143 ).
Q13: The unit of ki: no minus, so revise the units in the whole manuscript.
Reply: Related errors have been corrected in the revised manuscript and highlighted in yellow (Line 179 ).
Q14: Line 175: non-linear.
Reply: Related errors have been corrected in the revised manuscript and highlighted in yellow (Line 181).
Q15: Figure 6. Correct freundich to Freundlich.
Reply: Thanks the reviewer’s comment, the related errors have been corrected in Figure 6.
Q16: Line: nF (F is a subscript).
Reply: Related errors have been corrected in the revised manuscript and highlighted in yellow (Line 248).
Q17: Lines 208-210: the statement may need more discussion and to be supported with reference(s).
Reply: Thanks the reviewer’s comment, we have added two new references, and the more discussions are supported in the revised manuscript and highlighted in yellow (Line 218~221).
Q18: Line 243: correct English (It shows).
Reply: Related errors have been corrected in the revised manuscript and highlighted in yellow (Line 255~256).
Q19: In table 4: capitalize the first letter (tamarind, corncobs).
Reply: In Table 4, Related errors have been corrected in the revised manuscript and highlighted in yellow .
Q20: Line 251: Kd instead of K.
Reply: Related errors have been corrected in the revised manuscript and highlighted in yellow (Line 264).
Q21: Section 3.1.: consider reporting purity for all chemicals.
Reply: Related errors have been corrected in the revised manuscript and highlighted in yellow .(Section 3.1).
Q22: Line 285: referring to KOH mass make no sense because its purity was not reported.
Reply: Related errors have been corrected in the revised manuscript and highlighted in yellow .(Section 3.1).
Q23: What is the idea behind selection of the NC precursors ratio as corncon:KOH: Fe3+ of 1:1.87:2.5 M Fe3+. Is it based on any criteria or is it optimized first or performed according to a previous, justified method?
Reply: Following the reviewer’s comment, we have enriched the relevant literature[ ref.35] and revised in the revised manuscript and highlighted in yellow.
Q24: Line 288: report the N2 gas flow rate.
Reply: we have been corrected in the revised manuscript and highlighted in yellow .(Line 300-303).
Q25: Line 311: pore volume of …(of what?).
Reply: Related errors have been corrected in the revised manuscript and highlighted in yellow .(Line 325).
Q26: Lines 343-344: the sentence is not clear.
Reply: Related sentences have been corrected in the revised manuscript and highlighted in yellow .(Line 359-360).
Q27: Line 344: 25°C or 25 °C, keep one style.
Reply: Related errors have been corrected in the revised manuscript and highlighted in yellow .
Q28: Line 345: add space (to be 12 h).
Reply: Related errors have been corrected in the revised manuscript.
Q29: Line 345: it is important to report the HNO3 volume used in each cycle.
Reply: Related sentences have been corrected in the revised manuscript and highlighted in yellow .(Line 361-362).
Q30: Used one style to abbreviate composite. Is MCB/ALG differs from MCB-ALG.
Reply: Related errors have been corrected in the revised manuscript.
Q31: Line 376: no XPS test in the report?
Reply: Related errors have been corrected in the revised manuscript.(see section Acknowledgments )
Thanks for your comments and suggestions on our manuscript!
Reviewer 3 Report
In this manuscript by Liu et al, the authors present the synthesis of corncob derived biochar alginate beads modified with KOH/FeCl3 and their performance in removal of methylene blue (MB) from contaminated water via adsorption. The beads have been characterized with nitrogen adsorption, microscopy, FTIR and XRD, while their performance as adsorbents for MB from waste water has been thoroughly studied by collecting adsorption data at three temperatures as a function of the pH and equilibrium MB concentration. Kinetics of adsorption has been modeled to reveal the mechanism of adsorption. The synthesized samples show impressive adsorption capacity. Further, they exhibit good re-usability for MB removal even after 5 cycles. This work is important and seems to have been carried out systematically and thoroughly. It is therefore suitable for publication in Molecules. However, I have some minor comments related to presentation, listed below, that the authors should address before publication
1. There are several linguistic errors throughout the manuscript. While most of these are benign and the reader can still understand the content, at some places these errors twist the meaning of what the authors actually intend to say. For example, in the title itself, the authors state 'Adsorption performance of methylene blue by ...'. This suggests evaluating the (adsorption) performance of MB whereas in the work the authors are evaluating the performance of the adsorbent (MCB/ALG) for removing MB from wastewater. Similarly at line 10 in the abstract, '...were tested for adsorption in MB dye wastewater...' should rather be '...were tested for adsorption of MB dye from wastewater...'. The authors should correct such mistakes.
2. The SEM images shown in Figure 1 have different length scales and so it is not easy to compare them. The authors should show all images with the same length scale and if needed they can show additional zoomed in counterparts of some images to show features at smaller length scales.
3. A sentences is repeated between lines 116 and 118.
4. In the XRD spectra shown in Figure 3, what are the peaks at 28.3 and 40.5 in the red curve?
5. At line 163, the authors probably mean 'time evolution' instead of 'time distribution' which has a different meaning.
6. Sample nomenclature is confusing. For example, the methods section lists three MCB/ALG samples - MCB/ALG-1, MCB/ALG-2, MCB-ALG3. What is the difference between these? Are they just three different batches of the same sample, or do they differ in SA content, as the results section seems to suggest? What are the three samples shown in Figure 4? Are they MCB/ALG-n (n=1,2,3) that have been mislabeled as MCB-n?
7. The authors begin the results section by using the abbreviated form of sample names whereas their full forms are given much later in the methods section. Full forms of abbreviations should be provided at first use in the paper.
Author Response
Reviewer #3:
Q1: There are several linguistic errors throughout the manuscript. While most of these are benign and the reader can still understand the content, at some places these errors twist the meaning of what the authors actually intend to say. For example, in the title itself, the authors state 'Adsorption performance of methylene blue by ...'. This suggests evaluating the (adsorption) performance of MB whereas in the work the authors are evaluating the performance of the adsorbent (MCB/ALG) for removing MB from wastewater. Similarly at line 10 in the abstract, '...were tested for adsorption in MB dye wastewater...' should rather be '...were tested for adsorption of MB dye from wastewater...'. The authors should correct such mistakes.
Reply: Thanks the reviewer’s comment, the related errors have been corrected carefully in the revised manuscript, such as “The prepared MCB/ALG beads were tested for adsorption of methylene blue (MB) dye from wastewater” and so on .
Q2: The SEM images shown in Figure 1 have different length scales and so it is not easy to compare them. The authors should show all images with the same length scale and if needed they can show additional zoomed in counterparts of some images to show features at smaller length scales.
Reply: Thanks the reviewer’s comment, relevant dimensions have been corrected in the manuscript in Fig1.
Q3: A sentences is repeated between lines 116 and 118.
Reply: Thanks the reviewer’s comment, the relevant sentence have been corrected in the manuscript in Line 119 and 120.
Q4: In the XRD spectra shown in Figure 3, what are the peaks at 28.3 and 40.5 in the red curve?
Reply: Thanks the reviewer’s comment, the peaks at 28.3 and 40.5 in the red curve may be the miscellaneous peak of corncob biochar, which we have confirmed in relevant literature. The two peaks are not marked in the revised manuscript,see Fig. 3.
Q5: At line 163, the authors probably mean 'time evolution' instead of 'time distribution' which has a different meaning.
Reply: Thanks the reviewer’s comment, related errors have been corrected in the revised manuscript ( see line 169)
Q6: Sample nomenclature is confusing. For example, the methods section lists three MCB/ALG samples - MCB/ALG-1, MCB/ALG-2, MCB-ALG3. What is the difference between these?
Reply: We have renamed the names of the sample and named it MCB/ALG-1, MCB/ALG-2 and MCB/ALG-3 respectively, according to the mass ratio of MCB and SA in the composite beads. Here MCB/ALG-1 means the mass ratio of MCB and SA in the composite bead is 1; MCB/ALG-2 means the mass ratio of MCB and SA in the composite bead is 2, and MCB/ALG-3 means the mass ratio of MCB and SA in the composite bead is 3 in turn. The difference between the three MCB/ALG samples is the proportion(SA content) of the added SA in the MCB.
Are they just three different batches of the same sample, or do they differ in SA content, as the results section seems to suggest?
Reply: No, the difference between the three MCB/ALG samples is the proportion(SA content) of the added SA in the MCB.
What are the three samples shown in Figure 4? Are they MCB/ALG-n (n=1,2,3) that have been mislabeled as MCB-n?
Reply: Fig. 4a is MCB/ALG-1, MCB/ALG-2 and MCB/ALG-3, respectively. Fig 4b is MCB/ALG-1. No, MCB/ALG-n (n=1,2,3) are described in the in the revised manuscript.(See Line 316-318).
Q7: The authors begin the results section by using the abbreviated form of sample names whereas their full forms are given much later in the methods section. Full forms of abbreviations should be provided at first use in the paper.
Reply: According to the reviewer’s comments, all the full forms of abbreviations are be provided at first use in the paper.
Thanks for your comments and suggestions on our manuscript!
Round 2
Reviewer 1 Report
The authors have modified the MS properly according to the comments,and it can be published.
Author Response
Thanks very much for your kind work and consideration on publication of our paper.
Reviewer 2 Report
The authors have answered my questions, and the manuscript is now suitable for publication.
The following minor points have to be considered as well.
- Q5: The MCB-1 still present in the Figure caption?
- Q10: I am sorry, there was misunderstanding. I meant the minus which is minus not dash or hyphen like the one in cm-1 (should be cm−1), and so on.
- Q19: The authors did not capitalized corncobs. If it is treated as a special term you can leave it without capitalization of the first letter. If not, it is better to capitalize the first letter for consistency.
- Q23: You did correct the temperature used for pyrolysis from 850 to 800 °C in line 295. But it is still reported in Table 4 as 850 °C. Please revise it again.
No more comments.
Author Response
Round 2 : Response to Reviewers
Reviewer #2:
Q5: The MCB-1 still present in the Figure caption?
Reply: Following the reviewer’s comment, we have checked the error. The revised manuscript has been corrected in Fig. 4 and highlighted in yellow (see Line 165-166). Thanks!
- Q10: I am sorry, there was misunderstanding. I meant the minus which is minus not dash or hyphen like the one in cm-1 (should be cm−1), and so on.
Reply: Thank the reviewer’s comment, we have checked carefully, the errors in the manuscript have been revised and highlighted in yellow (see Line 116, 120-122). Thanks.
Q19: The authors did not capitalized corncobs. If it is treated as a special term you can leave it without capitalization of the first letter. If not, it is better to capitalize the first letter for consistency.
Reply: Thank the reviewer’s comment, related errors have been corrected in the revised manuscript and highlighted in yellow (see Table 4).
Q23: You did correct the temperature used for pyrolysis from 850 to 800 °C in line 295. But it is still reported in Table 4 as 850 °C. Please revise it again.
Reply: Thank the reviewer’s comment, we have checked carefully, the errors in the manuscript have been revised and highlighted in yellow (see Table 4).
Thanks.